# Hyperreflective Dots on SD-OCT: Implications for Predicting Treatment Outcomes in Diabetic Macular Edema

**DOI:** 10.3390/diagnostics15192539

**Published:** 2025-10-09

**Authors:** Siying Li, Muzi Li, Aimin Sun, Hongwei Zhang

**Affiliations:** 1Department of Ophthalmology, Peking University People’s Hospital, Beijing 100044, China; 2Beijing Key Laboratory of Ocular Disease and Optometry Science, Peking University People’s Hospital, Beijing 100044, China; 3Department of Ophthalmology, The First Hospital of China Medical University, Shenyang 110001, China

**Keywords:** hyperreflective dots, spectral-domain optical coherence tomography, diabetic macular edema, ranibizumab, dexamethasone

## Abstract

**Objectives:** To evaluate the relationship between hyperreflective dots (HRDs) observed on spectral-domain optical coherence tomography (SD-OCT) and the outcomes following treatment with intravitreal ranibizumab or dexamethasone injections in patients with diabetic macular edema (DME). **Methods:** This retrospective study focused on individuals suffering from diabetic macular edema (DME) who underwent a sequence of three intravitreal ranibizumab injections. Based on treatment response, the eyes were categorized into two groups: responders and non-responders. The non-responder group subsequently received intravitreal dexamethasone (IVO) implants. Treatment results were evaluated by changes in BCVA, HRD number, and central macular thickness (CMT). **Results:** This research involved 112 eyes from 78 participants who had been diagnosed with DME. Seventy-three eyes (65%) were identified as ranibizumab responders and 39 eyes (35%) as ranibizumab non-responders. Of the 39 individuals who had suboptimal response to ranibizumab and subsequently received treatment with an intravitreal dexamethasone implant, 26 eyes (66.67%) exhibited a favorable response, while 13 eyes (33.33%) showed an insufficient response. IVR responders demonstrated significantly greater improvements in BCVA (0.54 ± 0.73 to 0.35 ± 0.40 logMAR vs. 0.52 ± 0.61 to 0.47 ± 0.38 logMAR) and CMT (456.53 ± 109.73 μm to 235.47 ± 49.13 μm vs. 468.99 ± 127.10 μm to 427.45 ± 52.91 μm) reduction. Baseline analysis revealed IVR non-responders had higher counts of both inner and outer retinal HRDs compared to responders (9.09 ± 3.38 vs. 7.07 ± 2.32 and 5.46 ± 2.03 vs. 4.27 ± 1.87, *p* < 0.05, respectively). Eyes with initially higher numbers of inner retinal HRDs, outer retinal HRDs, and subretinal HRDs demonstrated a significantly enhanced response to dexamethasone therapy (9.03 ± 3.18 vs. 7.55 ± 2.72, 6.55 ± 2.46 vs. 4.79 ± 1.88 and 0.27 ± 0.54 vs. 0.21 ± 0.47, *p* < 0.05, respectively). **Conclusions:** HRDs could potentially be used as a predictive biomarker to assess the effectiveness of anti-VEGF therapy in treating DME. Patients exhibiting a greater number of retinal HRDs tend to have less favorable reactions to anti-VEGF treatments but experience improved results with dexamethasone.

## 1. Introduction

Diabetic macular edema (DME) stands as a severe microvascular complication arising from diabetic retinopathy, and it is widely recognized as the primary cause of vision impairment associated with diabetes mellitus [1]. The incidence of DME exhibits notable variability across different patient populations: among individuals with type 1 diabetes, the rate ranges significantly from 4.2% to 14.3%, while in those with type 2 diabetes, the incidence is distinctly lower, falling within the range of 1.4% to 5.57% [2]. This discrepancy in incidence rates underscores the need for tailored approaches to understanding and managing DME across different diabetes subtypes.

A defining feature of diabetic macular edema (DME) is retinal thickening, which arises from the abnormal accumulation of fluid in the macular region due to impaired retinal vasculature. The underlying pathophysiology begins with long-term exposure to hyperglycemia, which exerts detrimental effects on retinal endothelial cells and the basement membrane. This damage leads to a cascade of events, including increased vascular hyperpermeability, the release of proinflammatory cytokines, and the development of retinal ischemia. In response to this ischemic microenvironment, there is a pathological overexpression of vascular endothelial growth factor (VEGF) and other pro-angiogenic factors [3]. This molecular cascade forms the basis for the use of intravitreal anti-VEGF therapies, which target VEGF to mitigate vascular leakage, reduce edema, and ultimately improve visual function. However, given the heterogeneity of DME pathology among individual patients, alternative treatment strategies, such as intravitreal steroid therapy, are also employed in clinical practice to address cases where anti-VEGF agents may be less effective [4,5].

Retinal hyperreflective dots (HRDs) manifest as unique, dot-like lesions measuring from 20 to 40 μm in diameter, showing reflectivity that equal to or surpasses the retinal pigment epithelium (RPE) band seen on spectral-domain optical coherence tomography (SD-OCT). Although their precise origin is not well-defined, HRDs are associated with lipoprotein extravasation or inflammatory processes in the retinal [6]. Recent research suggests that HRDs are linked to the visual clarity outcomes following anti-VEGF therapy for macular edema [7]. Consequently, we performed a retrospective analysis on patients with diabetic macular edema who underwent three protocol-defined intravitreal ranibizumab injections (IVR). The study aimed to explore the association between HRDs on SD-OCT and treatment outcomes including the reduction in central macular thickness (CMT) and enhancement of best corrected visual acuity (BCVA), among both those who responded to anti-VEGF and those who did not, then switched to dexamethasone implants.

## 2. Methods

### 2.1. Patients

This retrospective cohort study evaluated patients with diabetic macular edema (DME) treated at Peking University People’s Hospital (November 2023–April 2025) who received intravitreal injections of ranibizumab (ranibizumab, Lucentis; Novartis, Pharma AG, Zug, Switzerland) as well as who switched to ozurdex (Ozurdex; Allergan, Inc., Irvine, CA, USA) implants yielding suboptimal response after three IVR injections. We conducted a quantitative comparison of hyperreflective dot (HRD) counts on spectral-domain OCT between treatment responders and non-responders. The study adhered to the tenets of the Declaration of Helsinki and was approved by the Institutional Review Board (or Ethics Committee) of Peking University People’s Hospital (approval code 2023PHD005-001, approval date 1 May 2025). The written informed consent was obtained from all individual participants included in the study.

The inclusion criteria were as follows: (1) adults 18 years or older diagnosed with type II diabetes; (2) The thickness of the central macula is 300 μm or more; (3) treatment-naïve DME initially managed with IVR. The exclusion criteria were as follows: (1) individuals with high myopia (>8 diopters) resulting in low-quality SD-OCT scan quality); (2) Opaque refractive media (cataract, corneal disease, vitreous hemorrhage and so on); (3) eyes affected by any ocular disease that could obstruct visual enhancement other than DME, such as optic nerve diseases or macular hole; (4) Eyes that have undergone any intraocular treatment, including surgery, laser therapy, and so on; (5) patients with uncontrolled systemic diseases or infectious diseases.

### 2.2. Examination

All subjects underwent completed a comprehensive initial eye examination, which included best-corrected visual acuity (BCVA), intraocular pressure (IOP), slit-lamp biomicroscopy, indirect ophthalmoscopy, color fundus photography, spectral-domain optical coherence tomography (SD-OCT) (CIRRUS HD-OCT Model 5000, Carl Zeiss Meditec, Jena, Germany) as well as ultra-wide field (UWF) fundus fluorescein angiography (FFA) using Optos 200Tx (Optos plc, Dunfermline, UK). Subsequent visits included BCVA, IOP, evaluations of the anterior and posterior segments, and SD-OCT. For statistical evaluation, BCVA was transformed into the logMAR scale, which represents the logarithmic value of the minimal angle of resolution. A vision of counting fingers was measured as 2.0 logMAR, while hand movement was recorded as 3.0 logMAR. The measurement of central macular thickness (CMT) utilized the device’s follow-up tracking capability, maintaining consistent scan positioning at identical macular locations during every visit through horizontal and vertical B-scan alignment.

### 2.3. Treatment

All the patients enrolled in the study received a standardized treatment regimen involving monthly intravitreal injections of 0.05 mL ranibizumab, administered consistently over a three-month period. Following the completion of this three-injection course, the eyes were categorized based on their therapeutic response to intravitreal ranibizumab (IVR) treatment. Specifically, patients were classified as ranibizumab responders if they demonstrated a positive treatment response, which was defined by meeting one of two criteria: either achieving a central macular thickness (CMT) of less than 300 µm, or experiencing a reduction in CMT of more than 50 µm after the three consecutive IVR injections.

For those patients whose condition did not show an adequate response to IVR—meaning they failed to meet the aforementioned response criteria—dexamethasone implants were administered as a subsequent therapeutic intervention. Then, one month after the dexamethasone implant was delivered, the patients who had initially been non-responsive to ranibizumab underwent a follow-up assessment to evaluate their response to the dexamethasone therapy.

### 2.4. HFs Counting Methods

HRDs (hyperreflective dots) were characterized in this study as distinct, well-defined particulate structures with a diameter ranging specifically between 20 and 40 μm. A key identifying feature of these dots is their reflectivity on spectral-domain optical coherence tomography (SD-OCT) scans, which is either comparable to, or exceeds, the reflectivity of the retinal pigment epithelium (RPE) band-a consistent reference point in retinal imaging. For each patient involved in the study, a horizontal B-scan using SD-OCT was acquired across the foveal region at two critical time points: during the initial baseline visit and at subsequent follow-up consultations. These scans were then analyzed by an experienced grader (QJF) who remained masked to patient outcomes, ensuring objective assessment. The grader quantified the number of hyperreflective dots within the central 1500 um area of the fovea. Hyperreflective dots (HRDs) were categorized into three distinct groups based on their precise anatomical location within the retinal layers: (1) Inner retinal layers, extending from the internal limiting membrane (ILM) to the outer nuclear layer (ONL); (2) Outer retinal layers, located between the external limiting membrane (ELM) and the photoreceptors; (3) Subretinal region, situated from the neuro-retinal layer to the retinal pigment epithelium (RPE).

### 2.5. Statistical Analysis

Initial patient details were gathered and examined with SPSS Statistics version 19.0 software (IBM SPSS Inc., Chicago, IL, USA). For datasets that deviate from a normal distribution, the continuous variables are depicted using median values and interquartile ranges. For analyzing normally distributed datasets, Student’s *t*-test was applied, whereas the Wilcoxon signed-rank test was used for evaluating nonparametric datasets. Categorical variables were evaluated using Pearson’s chi-squared test or Fisher’s exact test, with statistical significance defined as *p* < 0.05.

## 3. Results

### 3.1. Baseline Demographics

In this study, 112 eyes from 78 patients diagnosed with diabetic macular edema (DME) were included for analysis. Among these, 73 eyes (65%) demonstrated a positive response to ranibizumab treatment, as determined by predefined clinical criteria such as improved best-corrected visual acuity (BCVA) and reduced central retinal thickness (CRT) (Figure 1). In contrast, 39 eyes (35%) did not show a well-defined response, failing to meet these key improvement benchmarks. The 39 eyes that responded poorly to ranibizumab were subsequently treated with an intravitreal dexamethasone implant. Following this intervention, 26 eyes (66.67%) achieved a favorable outcome (Figure 2), while 13 eyes (33.33%) continued to exhibit a poor response to the treatment (Figure 3).

Table 1 and Table 2 present detailed baseline characteristics and demographic data for both responders and non-responders to ranibizumab (IVR) and dexamethasone implant (IVO) treatments. A comprehensive statistical analysis was conducted, comparing various parameters across the groups. No significant differences were detected between IVR responders and non-responders, nor between IVO responders and non-responders, in terms of demographic characteristics (age, sex, BMI), general health status (HbA1c level, diabetes duration, presence of hypertension), ocular history (lens status, prior laser photocoagulation), or baseline ocular features (DRILs, EZ/ELM disruption, BCVA, IOP, CRT).

### 3.2. Outcomes

The IVR responder group exhibited a significant improvement in both BCVA (best-corrected visual acuity) and CMT (central macular thickness), as detailed in Table 3 and Table 4. Specifically, responders showed a more pronounced enhancement in BCVA compared to non-responders: their measurements shifted from 0.54 ± 0.73 to 0.35 ± 0.40 logMAR, whereas non-responders only improved from 0.52 ± 0.61 to 0.47 ± 0.38 logMAR, with this difference reaching statistical significance (*p* < 0.05). Similarly, the reduction in CMT was far more substantial among responders, dropping from 456.53 ± 109.73 μm to 235.47 ± 49.13 μm, in contrast to non-responders whose CMT decreased to a much lesser extent (from 468.99 ± 127.10 μm to 427.45 ± 52.91 μm), and this disparity was also statistically significant (*p* < 0.05).

However, the changes in HRDs (hyperreflective dots) followed a different pattern. At the initial assessment, patients who did not respond to ranibizumab had significantly higher counts of both inner and outer retinal HRDs than those who responded well: inner retinal HRDs were 9.09 ± 3.38 versus 7.07 ± 2.32, and outer retinal HRDs were 5.46 ± 2.03 versus 4.27 ± 1.87, with both differences significant at *p* < 0.05. In contrast, there was no notable variance in subretinal HRDs between the two groups at baseline. After IVR (intravitreal ranibizumab) injection, responders experienced a marked decrease in inner retinal HRDs, falling from 7.07 ± 2.32 to 3.51 ± 1.78, while non-responders showed no such decline (remaining from 9.09 ± 3.38 to 8.21 ± 2.21), a difference that was statistically significant (*p* < 0.05). Meanwhile, both groups saw a significant reduction in subretinal HRDs, though the decrease in outer retinal HRDs did not reach statistical significance in either cohort.

A positive correlation was identified between the reduction in total HRDs and improvements in BCVA (r = 0.69, *p* < 0.001), meaning a more substantial decrease in total HRDs was associated with greater gains in visual acuity. Similarly, a positive relationship was observed between HRD decline and CMT reduction (r = 0.63, *p* < 0.001), indicating that a larger decrease in HRDs corresponded with more pronounced central macular thinning.

In contrast to the response pattern seen with ranibizumab, eyes with higher initial counts of inner, outer, and subretinal HRDs demonstrated a significantly better response to dexamethasone therapy: inner retinal HRDs were 9.03 ± 3.18 versus 7.55 ± 2.72, outer retinal HRDs were 6.55 ± 2.46 versus 4.79 ± 1.88, and subretinal HRDs were 0.27 ± 0.54 versus 0.21 ± 0.47, with all *p*-values < 0.05, as shown in Table 5 and Table 6. Additionally, ozurdex responders had significantly greater reductions in HRDs across inner, outer, and subretinal layers compared to non-responders: inner retinal HRDs dropped from 9.03 ± 3.18 to 4.31 ± 2.72 versus 7.55 ± 2.72 to 5.75 ± 2.99, outer retinal HRDs from 6.55 ± 2.46 to 3.49 ± 1.84 versus 4.79 ± 1.88 to 3.67 ± 1.88, and subretinal HRDs from 0.27 ± 0.54 to 0.13 ± 0.39 versus 0.21 ± 0.47 to 0.18 ± 0.27, with all differences significant at *p* < 0.05.

## 4. Discussion

This study comprehensively evaluated the relationship between hyperreflective dots (HRDs) detected by spectral–domain optical coherence tomography (SD-OCT) and the efficacy of intravitreal ranibizumab (IVR) or dexamethasone implants in the treatment of diabetic macular edema (DME). The results demonstrated that cases with poor responses to ranibizumab often had a higher number of baseline HRDs, while better therapeutic outcomes were achieved with dexamethasone treatment. Moreover, the degree of HRD resolution was positively correlated with improvements in best-corrected visual acuity (BCVA) and reductions in central macular thickness (CMT), highlighting the potential of HRDs as a predictive biomarker in DME management.

Discrete and localized lesions, known as hyperreflective dots (HRDs), have been observed in several retinal conditions such as age-related macular degeneration (AMD), retinal vein occlusion (RVO), central serous chorioretinopathy, Stargardt disease, retinitis pigmentosa, and diabetic retinopathy [8,9,10,11]. Despite their frequent observation, the pathophysiological basis of HRDs remains incompletely understood. To date, several competing theories have emerged to explain the formation of hyperreflective dots (HRDs), with proposed mechanisms ranging from lipoprotein extravasation and inflammatory processes to microglial activation. Bolz and Chen et al. identified HRDs as morphological correlates of lipid extravasation in DME [12,13]. Their research highlighted that these well-defined hyperreflective structures tend to localize within the walls of retinal microaneurysms and distribute across multiple retinal layers. Supported by histological analyses, the authors suggested that these foci likely represent subclinical lipoprotein leakage, a consequence of breakdown in the inner blood-retinal barrier (BRB)—a critical structure that regulates the passage of substances between retinal blood vessels and the neurosensory retina. Building on this, Bolz and colleagues proposed that HRDs may serve as early markers of BRB compromise in DME, reflecting the initial extravasation of lipoproteins and proteins before more overt signs of edema or vascular dysfunction become apparent. Similarly, Esther et al. believe that the distribution of HRDs is related to the intraretinal fluid (IRF), representing the accumulation of fluid within the retina caused by the disruption of the blood-retinal barrier [14]. Their histopathological investigations further indicated that these localized hyperreflective changes are likely subclinical manifestations of lipoprotein leakage driven by inner BRB disruption. From this correlation, the authors inferred that HRDs signify the escape of lipoproteins and/or plasma proteins into the retinal parenchyma, positioning them as preclinical markers of barrier dysfunction in the early stages of diabetic macular edema. Together, these studies underscore the potential of HRDs to serve as sensitive indicators of underlying vascular compromise, offering insights into the early pathophysiology of DME and other retinal disorders.

Conversely, various alternative studies have suggested a connection between HRDs and inflammatory responses in the retina [15,16]. Inflammation in the retina activates microglia, causing cell swelling and migration through the retinal layers, which eventually results in the formation of hyperreflective dots (HRDs) in age-related macular degeneration (AMD). Progressive retinal inflammation activates microglial cells, triggering their proliferation, migration across retinal layers, and morphological changes leading to aggregation. Vujosevic et al. observed that HRDs initially localize to the microglia-rich inner retina in early DR, later extending to outer layers as retinopathy progresses [17]. These findings linked outer retinal HRDs to photoreceptor layer damage and visual acuity impairment in DME, suggesting that HRDs may reflect microglial activation, with the aggregated, activated cells appearing as hyperreflective dots on SD-OCT. Previous studies have documented the movement of HRDs from the inner to outer retina over the course of DR progression [18]. This may reflect a dynamic interplay between BRB disruption and inflammatory activation, with their distribution and quantity varying based on disease stage and underlying pathology [19].

In the complex pathophysiology of DME, elevated vascular endothelial growth factor (VEGF) levels, which are induced by factors such as chronic hyperglycemia and retinal hypoxia, play a pivotal role [20]. VEGF is considered the primary agent responsible for the progression of diabetic macular edema (DME). VEGF promotes increased vascular permeability in the retinal microvasculature. This occurs through multiple mechanisms, including the activation of endothelial cell signaling pathways that lead to the loosening of tight junctions between endothelial cells in the blood–retinal barrier. As a result, plasma components, including proteins and fluid, leak into the extracellular space of the retina, ultimately causing the characteristic macular edema. Anti-VEGF therapies have emerged as a cornerstone in the treatment of DME [21]. Among these, ranibizumab is a well- studied agent. Ranibizumab is a humanized monoclonal antibody fragment that binds with high affinity to all active isoforms of VEGF-A. By blocking VEGF-A, it inhibits the VEGF-mediated signaling cascade in endothelial cells, targeting this pathway to reduce leakage and improve retinal vascular integrity [22,23]. Recent evidence indicates that such treatment not only reduces vessel leakage but also decreases the number of hyperreflective foci (HFs) in DME patients [24,25]. Consistent with prior research, our study observed a significant reduction in HRDs after the initial anti-VEGF treatment in patients with DME. This highlights the importance of vascular leakage and subsequent lipoprotein extravasation in the development of HRD in the progression of DME. The leakage of lipoproteins and other plasma components, which is driven by elevated VEGF-induced vascular permeability, likely contributes to the formation of HRDs. As anti-VEGF therapy effectively targets the root cause of this leakage [21], it leads to a subsequent decrease in HRDs. This not only has implications for understanding the pathophysiology of DME but also for predicting treatment response and visual outcomes in these patients.

Beyond VEGF-mediated mechanisms, inflammation plays a critical role in DME pathogenesis [26]. In addition to the increased expression of vascular endothelial growth factor (VEGF), inflammatory processes play a crucial role in the development of diabetic macular edema (DME) [27,28]. Considering this dual aspect of the disease, treatments therapies targeting inflammatory mediators such as intravitreal corticosteroids, particularly sustained-release dexamethasone implants, have proven to be an effective therapy for DME [29]. In our study, the application of dexamethasone in cases with poor anti-VEGF response can achieve a very good response, which supports the theory that HRDs may be linked to the inflammatory process. Previous research indicated that hyperreflective dots are associated with poorer visual results in macular edema caused by retinal vascular diseases [30,31]. This result is consistent with ours. Our research identified a direct relationship between the extent of HRD decrease and the level of enhancement in BCVA. Differing from the results reported by Liu et al., which focused on hyperreflective dots and visual outcomes, our research also examined the relationship between HRD quantity and treatment response to ranibizumab or dexamethasone implants in DME. The observed variation in therapeutic efficacy may depend on HRD number, locations, potentially reflecting their distinct SD-OCT origins. Considering HRDs are linked to microglial clusters, a reduction in HRD quantity observed through SD-OCT probably indicates reduced inflammation within the retina. Our research revealed that eyes with DME that responded inadequately to ranibizumab had a higher quantity of HRDs in comparison to those eyes that were responsive to IVR. After switching to dexamethasone implant treatment, significant clinical improvement was observed in 67% of the IVR-refractory cases. Interestingly, those who responded well to dexamethasone showed elevated HRD counts compared to those who did not. Considering that HRDs are indicative of inflammatory activity, these results imply that inflammatory pathways might have a more prominent impact than mechanisms driven by VEGF in the development of these DME conditions.

This research had some limitations: Initially, it was based on a single-center retrospective evaluation with a relatively limited sample size and short follow-up duration, which might reduce the generalizability of the findings. Moreover, the mechanistic interpretation of HRDs remains largely speculative, as the study relies solely on imaging findings without histopathological validation. Hence, future studies might involve a greater number of cases, extend over a longer period across multiple centers, and integrate advanced imaging biomarkers or molecular analyses to clarify the biological underpinnings of HRDs.

## 5. Conclusions

HRDs could be utilized as a forecasting biomarker for assessing the efficacy of anti-VEGF therapy in DME. Patients who have a higher number of retinal HRDs generally respond less effectively to anti-VEGF medications but tend to experience improved outcomes when treated with dexamethasone. Thus, dexamethasone implants could potentially serve as a more efficient therapy option for DME cases with substantial HRD accumulation.

## Figures and Tables

**Figure 1 diagnostics-15-02539-f001:**
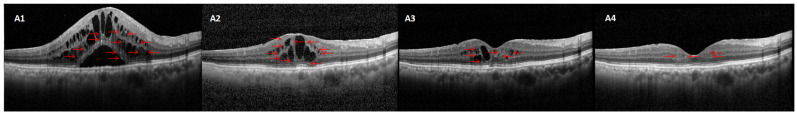
Example case of DME successfully treated with ranibizumab. The initial observation (**A1**) reveals a small number of hyperreflective dots (HRDs) (red arrow) alongside increased thickness of the central macular region (CMT). Post-injection evaluations (**A2**–**A4**) demonstrate a continuing decrease in CMT and HRDs.

**Figure 2 diagnostics-15-02539-f002:**
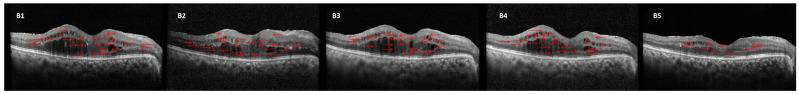
This case exemplifies DME with a positive response to dexamethasone and a lack of response to IVR treatment. Baseline (**B1**) and post-IVR (**B2**–**B4**) reveal a substantial number of HRDs (red arrow) with only slight CMT decline, indicating an IVR non-responder. One month administering dexamethasone (**B5**), CMT decreases notably with significant reduction in HRDs.

**Figure 3 diagnostics-15-02539-f003:**
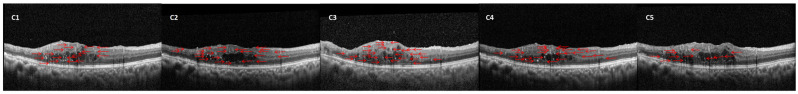
Example case of DME where dexamethasone treatment was ineffective. Baseline (**C1**) and post-IVR (**C2**–**C4**) showing abundant HRDs (red arrow) with persistent CMT elevation (IVR non-responder). After receiving dexamethasone (**C5**), there is no noticeable decrease in CMT, although the quantity of HRD has diminished.

**Table 1 diagnostics-15-02539-t001:** Comparison of initial characteristics between IVR responder and non-responder groups.

Characteristics	Responder Group (*n* = 73)	Non-Responder Group (*n* = 39)	*p* Value
Eyes/patients	73/52	39/26	-
Age (year), mean ± SD	59.27 ± 7.16	55.63 ± 7.73	0.776
Male/female	28/24	15/11	0.747
BMI (kg/m^2^), mean ± SD	24.88 ± 1.97	25.01 ± 1.73	0.706
Systemic profile:			
HbA1c (%), mean ± SD	6.66 ± 1.98	7.13 ± 1.87	0.273
Duration of DM (y), median (IQR)	8.35 (4.00, 11.50)	7.50 (4.50, 10.00)	0.234
Hypertension, *n* (%)	20 (38.5)	11 (42.3)	0.744
Ocular profile, *n* (%)			
History of laser	12 (16.4)	7 (17.9)	0.839
pseudophakic	9 (12.3)	6 (15.4)	0.651
DRIL, *n* (%)	2 (2.7)	2 (5.1)	0.516
ELM/EZ disruption, *n* (%)	3 (4.1)	2 (5.1)	0.804
LogMAR BCVA, mean ± SD	0.54 ± 0.73	0.52 ± 0.61	0.827
IOP (mmHg), mean ± SD	15.30 ± 3.47	14.83 ± 3.55	0.709
CMT (mm), mean ± SD	456.53 ± 109.73	468.99 ± 127.10	0.614

SD, standard deviation; BMI, Body Mass Index; IQR, interquartile range; DRIL, disorganization of retinal inner layers; ELM, external limiting membrane; EZ, ellipsoid zone; logMAR, logarithm of minimum angle of resolution; BCVA, best-corrected visual acuity; IOP, intraocular pressure; CMT, central macular thickness.

**Table 2 diagnostics-15-02539-t002:** Comparison of initial characteristics between IVO responder and non-responder groups.

Characteristics	Responder Group (*n* = 26)	Non-Responder Group (*n* = 13)	*p* Value
Eyes/patients	26/17	13/9	-
Age (year), mean ± SD	57.25 ± 7.36	54.58 ± 9.93	0.598
Male/female	10/7	5/4	0.873
BMI (kg/m^2^), mean ± SD	24.99 ± 2.73	25.11 ± 1.63	0.698
Systemic profile:			
HbA1c (%), mean ± SD	7.08 ± 1.17	7.19 ± 1.56	0.673
Duration of DM (y), median (IQR)	7.35 (4.25, 10.50)	7.80 (4.35, 11.00)	0.304
Hypertension, *n* (%)	7 (41.2)	4 (44.4)	0.873
Ocular profile, *n* (%)			
History of laser	5 (19.2)	2 (15.4)	0.768
pseudophakic	4 (15.4)	2 (15.4)	1.000
DRIL, *n* (%)	1 (3.8)	1 (7.7)	0.608
ELM/EZ disruption, *n* (%)	1 (3.8)	1 (7.7)	0.608
LogMAR BCVA, mean ± SD	0.53 ± 0.39	0.51 ± 0.67	0.772
IOP (mmHg), mean ± SD	14.60 ± 4.71	15.03 ± 2.59	0.609
CMT (mm), mean ± SD	472.53 ± 112.03	464.57 ± 137.83	0.511

SD, standard deviation; BMI, Body Mass Index; IQR, interquartile range; DRIL, disorganization of retinal inner layers; ELM, external limiting membrane; EZ, ellipsoid zone; logMAR, logarithm of minimum angle of resolution; BCVA, best-corrected visual acuity; IOP, intraocular pressure; CMT, central macular thickness.

**Table 3 diagnostics-15-02539-t003:** Comparison of BCVA at each visit between IVR responder and non-responder groups.

BCVA (LogMAR), Mean ± SD	Responder Group (*n* = 73)	Non-Responder Group (*n* = 39)	*p* Value
Baseline	0.54 ± 0.73	0.52 ± 0.61	0.827
1st injection	0.44 ± 0.42	0.50 ± 0.50	0.044 *
2nd injection	0.39 ± 0.37	0.49 ± 0.48	0.031 *
3rd injection	0.35 ± 0.40	0.47 ± 0.38	0.027 *

SD, standard deviation; logMAR, logarithm of minimum angle of resolution; BCVA, best-corrected visual acuity; *, statistically significant difference.

**Table 4 diagnostics-15-02539-t004:** Comparison of CMT at each visit between IVR responder and non-responder groups.

CMT (μm), Mean ± SD	Responder Group (*n* = 73)	Non-Responder Group (*n* = 39)	*p* Value
Baseline	456.53 ± 109.73	468.99 ± 127.10	0.614
1st injection	337.77 ± 93.18	440.95 ± 99.01	0.027 *
2nd injection	290.69 ± 66.38	433.18 ± 78.66	0.016 *
3rd injection	235.47 ± 49.13	427.45 ± 52.91	0.008 *

SD, standard deviation; CMT, central macular thickness; *, statistically significant difference.

**Table 5 diagnostics-15-02539-t005:** Comparison of BCVA between IVO responder and non-responder groups.

BCVA (LogMAR), Mean ± SD	Responder Group (*n* = 73)	Non-Responder Group (*n* = 39)	*p* Value
Baseline	0.53 ± 0.39	0.51 ± 0.67	0.772
1 month	0.37 ± 0.41	0.43 ± 0.88	0.039 *

SD, standard deviation; logMAR, logarithm of minimum angle of resolution; BCVA, best-corrected visual acuity; *, statistically significant difference.

**Table 6 diagnostics-15-02539-t006:** Comparison of CMT between IVO responder and non-responder groups.

CMT (μm), Mean ± SD	Responder Group (*n* = 73)	Non-Responder Group (*n* = 39)	*p* Value
Baseline	472.53 ± 112.03	464.57 ± 137.83	0.747
1 month	255.99 ± 69.83	397.48 ± 82.77	0.014 *

SD, standard deviation; CMT, central macular thickness; *, statistically significant difference.

## Data Availability

The original contributions presented in this study are included in the article. Further inquiries can be directed to the corresponding author.

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
