# Peer review of "Hyperreflective Dots on SD-OCT: Implications for Predicting Treatment Outcomes in Diabetic Macular Edema"

_diagnostics, 2025, doi:10.3390/diagnostics15192539_

Round 1

Reviewer 1 Report

Comments and Suggestions for Authors

The study addresses an important and clinically relevant question of  hyperreflective dots (HRDs) on SD-OCT  as predictive biomarkers for treatment outcomes in diabetic macular edema (DME).

The manuscript is clear, well-organized, well written, pleasant to read.  The results  align with recent literature on HRDs as indicators of both vascular and inflammatory pathology.

I suggest some minor revisions

  1. Abstract : rewrite the sentence " did not show well response" to " had suboptimal response"
  2. Terminology : The manuscript sometimes alternates between “hyperreflective dots” and “hyperreflective foci (HFs).” While HRDs and HRFs are related terms, they should be consistently defined and used throughout. They are synonymous 
  3. Responders were defined based on OCT in clinical practive. Please explain why BCVA was not included as the responder criteria 
  4. Figure 2 and " : arrow on HRD can be added for unfamiliar readers (non ophthalmologist)

Author Response

Dear Editor and Reviewers,

Thank you for your letter and for the time and effort that the reviewers put into concerning our manuscript.Those comments are all valuable and very helpful for revising and improving our paper, as well as the important guiding significance to our research. We have studied comments carefully and have made correction which we hope meet with approval. We have uploaded a revised manuscript and the revised portion are marked in the paper. Our point- by- point response to the reviewers’ comments are as followed:

Comments 1: Abstract : rewrite the sentence " did not show well response" to " had suboptimal response".

Response 1: Thank you for pointing this out. We have replaced the sentence " did not show well response" to " had suboptimal response" in line 22-23.

Comments 2: Terminology : The manuscript sometimes alternates between “hyperreflective dots” and “hyperreflective foci (HFs).” While HRDs and HRFs are related terms, they should be consistently defined and used throughout. They are synonymous.

Response 2: Thank you for pointing this out. We have standardized the terminology in our text, and "hyperreflective dots (HRDs)" is used consistently throughout the manuscript (line 136、line 266、line 296、line 303、line 341 and line 344) .

Comments 3: Responders were defined based on OCT in clinical practive. Please explain why BCVA was not included as the responder criteria.

Response 3: Thank you for your suggestion.Considering that BCVA (Best Corrected Visual Acuity) is affected by multiple factors, such as the status of the refractive media, the integrity of the ellipsoid zone (EZ), disorganization of the retinal inner layer, and age-related psychological factors. We chose to use the relatively objective CRT (Central Retinal Thickness) as the evaluation index. This ensures the uniformity and standardization of the measurement index.

Comments 4: Figure 2 and " : arrow on HRD can be added for unfamiliar readers (non ophthalmologist).

Response 4: Thank you for your suggestion.We have marked the HRDs with red arrows in the original figures to indicate the changes in the HRDs in Figure 1、Figure 2 and Figure 3.

Reviewer 2 Report

Comments and Suggestions for Authors

The article provides a valuable contribution by exploring the predictive role of hyperreflective dots (HRDs) on SD-OCT in determining treatment response to anti-VEGF versus dexamethasone in diabetic macular edema (DME). Its novelty lies in the comparative analysis of HRD distribution and outcomes across two therapeutic modalities, supporting the potential of HRDs as a biomarker for personalized treatment strategies. The structure is clear and follows the standard scientific format, progressing logically from introduction and methods through to results and discussion.

However, some limitations are evident. The retrospective single-center design, modest sample size, and relatively short follow-up period reduce the generalizability of the findings. While the study emphasizes statistical correlations, it does not fully address potential confounding factors such as systemic disease control or previous ocular treatments beyond the inclusion/exclusion criteria. Moreover, the mechanistic interpretation of HRDs remains largely speculative, as the study relies solely on imaging findings without histopathological validation.

Future work could be strengthened by multicenter, prospective trials with longer follow-up, as well as integration of advanced imaging biomarkers or molecular analyses to clarify the biological underpinnings of HRDs. Additionally, a more detailed exploration of how HRD-based stratification could be translated into clinical decision-making algorithms would further enhance the article’s impact.

Author Response

Dear Editor and Reviewers,

Thank you for your letter and for the time and effort that the reviewers put into concerning our manuscript.Those comments are all valuable and very helpful for revising and improving our paper, as well as the important guiding significance to our research. We have studied comments carefully and have made correction which we hope meet with approval. We have uploaded a revised manuscript and the revised portion are marked in the paper. Our point- by- point response to the reviewers’ comments are as followed:

Reviewer 2

The article provides a valuable contribution by exploring the predictive role of hyperreflective dots (HRDs) on SD-OCT in determining treatment response to anti-VEGF versus dexamethasone in diabetic macular edema (DME). Its novelty lies in the comparative analysis of HRD distribution and outcomes across two therapeutic modalities, supporting the potential of HRDs as a biomarker for personalized treatment strategies. The structure is clear and follows the standard scientific format, progressing logically from introduction and methods through to results and discussion.However, some limitations are evident. The retrospective single-center design, modest sample size, and relatively short follow-up period reduce the generalizability of the findings. While the study emphasizes statistical correlations, it does not fully address potential confounding factors such as systemic disease control or previous ocular treatments beyond the inclusion/exclusion criteria. Moreover, the mechanistic interpretation of HRDs remains largely speculative, as the study relies solely on imaging findings without histopathological validation. Future work could be strengthened by multicenter, prospective trials with longer follow-up, as well as integration of advanced imaging biomarkers or molecular analyses to clarify the biological underpinnings of HRDs. Additionally, a more detailed exploration of how HRD-based stratification could be translated into clinical decision-making algorithms would further enhance the article’s impact.

Thank you very much for your opinion.We have incorporated the limitations and future research directions you suggested into the manuscript, which has made our article more comprehensive (line 359-366). Additionally, we have refined the wording of the exclusion criteria (line 93-95). Specifically, patients who had received other intraocular treatments or had uncontrolled systemic diseases were indeed excluded from our study cohort. Thanks a lot again.
